# Factors Influencing Receipt and Type of Therapy Services in the NICU

**DOI:** 10.3390/bs13060481

**Published:** 2023-06-07

**Authors:** Christiana D. Butera, Shaaron E. Brown, Jennifer Burnsed, Jodi Darring, Amy D. Harper, Karen D. Hendricks-Muñoz, Megan Hyde, Audrey E. Kane, Meagan R. Miller, Richard D. Stevenson, Christine M. Spence, Leroy R. Thacker, Stacey C. Dusing

**Affiliations:** 1Division of Biokinesiology and Physical Therapy, University of Southern California, Los Angeles, CA 90033, USA; cbutera@usc.edu; 2Motor Development Lab, Department of Physical Therapy, Virginia Commonwealth University, Richmond, VA 23298, USAmrmiller2@vcu.edu (M.R.M.); 3Departments of Pediatrics and Neurology, Division of Neonatology, University of Virginia, Charlottesville, VA 22903, USA; jcw5b@hscmail.mcc.virginia.edu; 4Department of Pediatrics, Division of Neurodevelopmental and Behavioral Pediatrics, University of Virginia School of Medicine, Charlottesville, VA 22903, USA; 5Department of Neurology, Virginia Commonwealth University, Richmond, VA 23284, USA; 6Department of Pediatrics, Children’s Hospital of Richmond at VCU, Virginia Commonwealth University School of Medicine, Richmond, VA 23284, USA; karen.hendricks-munoz@vcuhealth.org (K.D.H.-M.);; 7Department of Physical Therapy, University of Virginia, Charlottesville, VA 22903, USA; mh5gt@hscmail.mcc.virginia.edu; 8Department of Counseling and Special Education, Virginia Commonwealth University, Richmond, VA 23284, USA; 9Department of Biostatistics, Virginia Commonwealth University School of Medicine, Richmond, VA 23284, USA

**Keywords:** therapy services, preterm infants, therapy frequency, neonatal intensive care unit

## Abstract

Understanding the type and frequency of current neonatal intensive care unit (NICU) therapy services and predictors of referral for therapy services is a crucial first step to supporting positive long-term outcomes in very preterm infants. This study enrolled 83 very preterm infants (<32 weeks, gestational age mean 26.5 ± 2.0 weeks; 38 male) from a longitudinal clinical trial. Race, neonatal medical index, neuroimaging, and frequency of therapy sessions were extracted from medical records. The Test of Infant Motor Performance and the General Movement Assessment were administered. Average weekly sessions of occupational therapy, physical therapy, and speech therapy were significantly different by type, but the magnitude and direction of the difference depended upon the discharge week. Infants at high risk for cerebral palsy based on their baseline General Movements Assessment scores received more therapy sessions than infants at low risk for cerebral palsy. Baseline General Movements Assessment was related to the mean number of occupational therapy sessions but not physical therapy or speech therapy sessions. Neonatal Medical Index scores and Test of Infant Motor Performance scores were not predictive of combined therapy services. Medical and developmental risk factors, as well as outcomes from therapy assessments, should be the basis for referral for therapy services in the neonatal intensive care unit.

## 1. Introduction

In 2020, the incidence of preterm birth (before 37 weeks of gestation) impacted one in every ten infants born in the United States [1]. Compared with term infants, the medical cost associated with preterm birth is almost doubled from birth to 2 years of age [2] and is highest for those with early preterm births [3]. Rates of preterm birth differ by race and ethnicity with the preterm birth rate at 14.4% among African American women, and 9.1% and 9.8% among white and Hispanic women, respectively [1]. Infants who are born preterm are at a significantly higher risk of developmental disability including cerebral palsy, intellectual disability, autism spectrum disorders, learning disabilities, and other general developmental delays [4,5,6,7]. These risks, along with attention deficit hyperactivity disorder (ADHD), brain injury, visual and hearing impairment, and cognitive impairment [8,9,10,11,12,13,14] lead to the need for specialized services such as physical therapy (PT), occupational therapy (OT), speech and language therapy services (ST), and special education services as well as many other medical subspecialty services [15]. Developmental difficulties can appear as early as term equivalent age, even before infants leave the Neonatal Intensive Care Unit (NICU) [16,17,18,19].

Therapies consisting of physical, occupational, or speech–language therapies in the NICU aim to improve the neurobehavioral, sensory, feeding behavior, state regulation, and neuromotor function of infants who were born less than 37 weeks of gestational age [20]. The American Academy of Pediatrics (AAP) recommends specialized OT, PT, and ST therapy services while infants are still in the NICU [21]. Each of these disciplines (OT, PT, and ST) has specific competency recommendations for training in order to provide specialized services to infants born preterm within the NICU [22,23,24]. There is evidence that therapy interventions beginning in the NICU have benefits on motor skills, oral motor skills, feeding volume, prevention of scapular–humeral tightness and shoulder retraction, exploratory problem-solving behaviors, and can result in less asymmetry of reflexes and movement [25,26,27,28,29,30]. Further, one systematic review demonstrates that parent-delivered motor interventions, as guided by a physical or occupational therapist, may improve both cognitive and motor outcomes in infants born preterm [30]. Engaging parents early throughout the NICU stay fosters relationship building between a therapist and parent, and provides parent education about the infant including how to developmentally support the infant [31]. This engagement is especially important towards the end of the NICU stay to support the transition from NICU to home.

Given the short and long-term developmental challenges associated with preterm birth, there is increased emphasis on training therapists to deliver interventions to preterm infants. Research on the effectiveness of targeted early interventions in the NICU is needed but should be considered when compared with the current standard of practice. Understanding the current state of therapy in the NICU, the type and amount of therapy being administered as usual care, and which demographic, behavioral, and medical risk factors are associated with access to therapy services will help with intervention research and public policy. Despite the importance of these questions, to the best of our knowledge, only one study to date has examined the type and frequency of therapy services provided for preterm infants in a single-level IV NICU [32]. They found that all included preterm infants in the NICU received OT and PT services, and 51% received ST [32]. Infants received OT, PT, and ST therapy an average of between one to two times per week for each service. Initial referral for PT or OT was due to positioning evaluation and intervention, then the routine continuation of therapy services was noted at 30 weeks of gestational age [32]. Sicker infants (those on respiratory supports, who had sepsis, or had a brain injury) received more therapy services before discharge and had an earlier initiation of OT and PT services. ST services were initiated at 36 weeks, coinciding with feeding/swallowing issues [32]. Though PT, OT, and ST services had some overlap in their interventions, there was a clear delineation between the services provided [32].

Here, we aim to add to this literature by using primary medical data that documented the frequency of therapy visits between baseline assessment and NICU discharge for infants born <32 weeks gestational age and who were part of the Supporting Play Exploration and Early Development Intervention (SPEEDI2) clinical trial (NCT03518736). SPEEDI is a three-arm randomized clinical trial, with participants enrolled at three sites. One arm (SPEEDI_Early) provides an intervention that starts in the NICU and aims to provide an enriched environment and increased opportunities for infant-initiated movements through collaborative parent, therapist, and infant interactions during the first months of life [33].

The objectives of this paper are to (1) describe the therapy services that very preterm infants received in the NICUs and evaluate if the frequency or type of services changed over time as infants moved closer to NICU discharge, and (2) evaluate if medical, behavioral, and sociodemographic infant risk factors (race, NMI, TIMP score, abnormal GMA) influenced amount or type of therapy services received in the NICU.

## 2. Methods

### 2.1. Recruitment and Consenting

Every infant admitted to participating Level IV NICUs during the enrollment period was screened for eligibility. Initially, only infants <29 weeks of gestation were enrolled from one of two hospitals; however, following the COVID-19 pandemic, the inclusion criteria were changed to <32 weeks of gestation and a NICU stay of greater than 28 days. These criteria are consistent with the state criteria for automatic eligibility for early intervention ensuring all infants were eligible for the same early intervention services. In addition, a community hospital with a level III NICU was added as an enrollment site, but all infants enrolled were counted toward the primary site’s enrollment as all study visits were completed by the primary hospital’s research team. Infants were offered enrollment if they were between 35 and 42 weeks of gestation, medically stable, off invasive or non-invasive ventilation, lived within 100 miles of the hospital, and spoke English. Exclusion criteria included a diagnosis of a genetic syndrome or musculoskeletal deformity.

### 2.2. Sample

Participants included 83 infants (mean gestational age of 26.5 (2) weeks) enrolled in a therapeutic clinical trial and were randomly assigned to one of 3 groups, Usual Care, SPEEDI_early, and SPEEDI_Late (NCT02153736). While only the infants in SPEEDI_Early were receiving NICU-based intervention visits, all 3 groups were monitored, and able to continue their usual clinical care including therapy services (Table 1). The usual care group received business-as-usual clinical care for the duration of the study, and the SPEEDI_Late group received additional intervention after being discharged from the NICU. The combined sample’s mean age at baseline assessment was 11.21 (3.57) weeks of chronological age or 37.35 (4.61) weeks of gestation (Table 1). More than 50 percent were considered to be at high risk for cerebral palsy or other neurodevelopmental disability based on having a brain injury demonstrated on cranial ultrasound or poor repertoire, cramped synchronized, or chaotic general movements at baseline. Of the participants, 46% percent were male, 52% were Caucasian, 31% were Black, and 14% identified as more than one race. Three percent identified as Hispanic (Table 1).

### 2.3. Primary Outcome Measures

All outcome measures used in this analysis were part of the research protocol and completed by highly trained research therapists. At baseline, the Test of Infant Motor Performance (TIMP), General Movement Assessment, and an initial medical record review were completed. Ongoing medical record reviews were completed by the site’s clinical research coordinator who was familiar with the site’s medical record system. Each hospital had similar policies for therapy documentation of any completed therapy visit.

Test of Infant Motor Performance (TIMP). The TIMP is an assessment of posture and movement for infants from 32 weeks of gestational age to 4 months of corrected age ([34] Campbell et al., 1995). Testing combines observation of spontaneous movements and placement in various positions to assess activities such as head centering, reaching, finger movements, and head and trunk control. The TIMP is a reliable and valid measure of motor performance [35] and is sensitive to age-related changes (r = 0.83) [36].

Prechtl’s Assessment of General Movements (GMA). The GMA is a standardized, noninvasive method of observation of spontaneous, complex movements to evaluate the typical maturation of the nervous system [37]. Prior studies have shown high sensitivity and specificity of the GMA to predict which children are at the highest risk for developing cerebral palsy (CP) [38,39]. Videos were scored for writhing movements by a certified and experienced investigator. Writhing movements were scored as normal or abnormal (poor repertoire, cramp synchronized, or chaotic). We used GMA classification to identify which infants were at high or low risk for CP for our risk strata variable.

Medical Record. Electronic medical records were reviewed weekly from the time of enrollment to NICU discharge. Weeks were considered Sunday to Saturday and thus the last Saturday the infant was in the NICU ended the final week of data extraction. If babies were given a baseline assessment and discharged within 24 h, their data were not included in the analysis. Data from full weeks were considered in all analyses. The length of NICU stay after the baseline assessment ranged from 0 days to more than 100 days (median 14 days, range 0–110 days). However, only 2 infants were still in the hospital 10 weeks after baseline; therefore, data provided in this paper are presented by week post baseline, including the infants who were in the hospital the entire week (Table 2). The frequency of sessions per therapy discipline was calculated and quantified based on the presence of a therapy note describing the usual care intervention session in the documentation. Neither site had a standard order set for therapy, so therapy visits were based on individual physician referral. Medical, behavioral, and sociodemographic infant risk factors (race, sex, NMI, and neuroimaging findings) were collected or calculated from the infant medical records and parent report surveys.

### 2.4. Statistics

Service Type Over Time. To describe the therapy services that very preterm infants received in the NICU, we fit a generalized linear model to the data for each week post baseline in which the infant was in the NICU, utilizing a Poisson distribution to model the mean number of services. The model included an effect for service Type (OT, PT, and ST), the number of weeks since baseline (1 to 9), and the interaction between Type and Week as well as a random effect for participant. We tested to see if we could treat Week as a continuous variable (with the built-in assumption of linearity), as opposed to treating Week as categorical (and thus a non-linear effect), and found that the less complex model using Week as a continuous variable was sufficient (likelihood ratio = 2.99, 21 d.f., *p* = 0.9999).

Service Type Before Discharge. To specifically explore the weeks leading up to hospital discharge, we fit a generalized linear model to the data for the three weeks prior to discharge, utilizing a Poisson distribution to model the mean number of services. The model included an effect for service Type (OT, PT, and ST), Discharge Relative Week (−3, −2, −1), and the interaction between Type and Discharge Relative Week as well as a random effect for participant. We tested to see if we could treat Week as a continuous variable (with the built-in assumption of linearity), as opposed to treating Week as categorical (and thus a non-linear effect), and found that similar to the above, the less complex model was sufficient (likelihood ratio = 2.48, 3 d.f., *p* = 0.4789).

Predictors of All Services. To evaluate if sociodemographic, neurological function, or medical risk factors, which can be measured by the medical team who make the referrals, influenced access to therapy in the NICU, we refit the initial service model (“Service Type Over Time”) described above and added in fixed effects for race (Caucasian yes/no), baseline NMI, and GMA (normal/abnormal).

Predictors of Individual Services. In order to determine if medical risk or standardized therapy assessment results influenced the frequency of individual therapy service, we repeated the analysis by adding the TIMP, which is typically completed by therapists and for each therapy service, rather than aggregate. We used three separate generalized linear models (baseline NMI, baseline TIMP, and baseline GMA) of the data for the frequency of therapy services from baseline through discharge or 9 weeks post baseline, whichever was shorter, utilizing a Poisson distribution to model the mean number of services. The models included a fixed effect for (NMI, TIMP, or GMA) as well as a random effect for participant. Parameter estimates were examined and post hoc tests performed with a Tukey HSD correction were calculated when relevant for pairwise comparisons (OT, PT, and ST).

## 3. Results

### 3.1. Service Type

We fit a generalized linear model including an effect for service Type (OT, PT, and ST), Week (one to nine), and the interaction between Type and Week as well as a random effect for participant. The interaction effect for Type by Week was significant (F_1,729_ = 7.6, *p* = 0.0005). This significant interaction indicates that services provided were significantly different, but the magnitude and significance of the difference depend upon the discharge week (Figure 1a.)

First, we focused only on the week of discharge and the three weeks before discharge to see if therapy services increased to get infants ready to be discharged (means and standard deviation, Table 3). In this model, the interaction effect for Type by Week was not significant (F_1,480_ = 0.06, *p* = 0.9428) and the model was refit to exclude the interaction effect. The Service Type effect was significant (F_2,482_ = 25.22, *p* < 0.0001) but the Week effect was not significant (F_1,482_ = 0.34, *p* = 0.5622). Post hoc tests revealed that ST services are significantly different from both OT (t_482_ = −5.61, adjusted *p* < 0.0001) and PT (t_482_ = −6.24, adjusted *p* < 0.0001) services, but there was no difference between OT and PT (t_482_ = 0.65, adjusted *p* = 0.7926) in terms of the frequency of services (Figure 1b).

### 3.2. Predictors of All Services—Medical, Race, and Severity Strata/GMA

In examining the significant parameter estimates, we found that when holding all things equal, infants at high risk for CP (abnormal GMA) received an average of 0.12 more therapy sessions than did infants at low risk for CP (Table 4). In addition, while holding all things equal, Caucasian infants received on average 0.11 fewer therapy sessions than did non-Caucasian infants. The medical risk was non-significant (Table 4).

### 3.3. Predictors of Individual Services—Severity Strata/GMA

Baseline GMA was related to the mean number of OT sessions over the 9 weeks of NICU time (t_54_ = 2.86, *p* = 0.006). Examination of the parameter estimate for GMA indicates that infants who had an abnormal baseline GMA had a higher mean number of OT sessions (0.51 more). The baseline GMA score was not statistically significantly associated with PT sessions (t_64_ = 0.08, *p* = 0.9356) or ST sessions (t_60_ = 0.02, *p* = 0.9827).

### 3.4. Predictors of Individual Services—Baseline NMI

Baseline NMI was related to the mean number of OT sessions over the nine weeks of NICU time (t_64_ = 3.19, *p* = 0.0022). Examination of the parameter estimate for NMI indicates that as the baseline NMI score increases, the mean number of OT sessions increases. With regards to the mean number of PT sessions, the baseline NMI score is not statistically significantly associated with PT sessions (t_64_ = −1.07, *p* = 0.2894). With regards to the mean number of ST sessions, we see that baseline NMI is related to the mean number of ST sessions over the 9 weeks of NICU time (t_889_ = −2.16, *p* = 0.0346). Examination of the parameter estimate for NMI indicates that as the baseline NMI score increases, the mean number of ST sessions decreases.

### 3.5. Predictors of Individual Services—Baseline TIMP

The baseline TIMP score was not associated with the mean number of OT sessions (t_44_ = −0.74, *p* = 0.4643), the mean number of PT sessions (t_53_ = 0.08, *p* = 0.9379), or the mean number of ST sessions (t_889_ = 0.03, *p* = 0.9795).

## 4. Discussion

This study found that services provided in the NICU were significantly different by type, but the magnitude and significance of the difference depended upon the time since enrollment week. Across weeks 1–9, OT and PT services increased over time, and ST decreased over time when the NICUs were combined. In weeks 1–6, infants received the most ST services, and in weeks 7–9 infants received the most OT services. Considering that the median length of stay was 14 days after the baseline assessment, the infants who remained in the NICU after 6 weeks were likely to have had a new onset of medical instability, prolonged feeding difficulty warranting placement of a G-tube, and reduced attempts at oral feeding, thus requiring less ST. However, the prolonged admission and increasing age are consistent with the need for more intervention focusing on social and play interaction that may have been provided by OT. Given the nature of the data used in this analysis, we are unable to determine the impact of medical stability and feeding outcomes.

In the three weeks leading up to discharge (Table 3), ST services were significantly greater than both OT and PT services, but there was no difference between OT and PT in the frequency of services. Infants at high risk for CP based on an abnormal GMA received more combined therapy sessions than did infants at low risk for CP. This relationship seemed largely driven by the mean number of OT sessions as the baseline GMA score was not statistically significantly associated with the number of PT or ST sessions. As the baseline NMI score increased, the mean number of OT sessions increased, PT sessions were not changed, and the mean number of ST sessions decreased. The baseline TIMP score was not associated with the mean number of OT, PT, or ST sessions. These findings must be considered within the context of the data, and NICU admission and staffing. The data reflect documented visits for clinical care. However, in acute care hospitals, the focus of therapy is often on supporting the discharge process for all patients resulting in staffing being pulled from the NICU to other areas to ensure discharge of older patients is not delayed. Thus, data on the planned or recommended therapeutic dose by the clinical care team are not included in this analysis; only data on the delivered sessions were analyzed. In addition, a portion of this study was completed during the COVID-19 pandemic, which influenced many aspects of care, from staffing to visitation in NICUs [40].

Our findings support the work of Ross et al. [32] by demonstrating that OTs, PTs, and STs have a role in providing therapeutic interventions early in gestation to high-risk infants in the NICU with concurrent medical interventions. The previous study found that sicker infants (those on respiratory supports, who had sepsis, or had a brain injury) received more therapy services before discharge and had an earlier initiation of OT and PT services. Our paper adds to this by showing that an abnormal GMA (a predictor of high risk for developing cerebral palsy) is also associated with more therapy services in the NICU and that NMI and TIMP scores are not within this sample. It should be noted that this paper [32] started collecting data earlier than the current study (30 weeks vs. 35 weeks of gestation).

Although infants at high risk for CP based on an abnormal GMA received more therapy sessions than did infants at low risk for CP, when looking at the service type, this relationship was only significant for the number of OT sessions. Similarly, the baseline NMI score predicted greater OT sessions only. The baseline TIMP score was not associated with the mean number of any type of therapy session. These results highlight the lack of valuable clinical and developmental information being used to guide therapy referrals. While GMA is well known as a strong predictor of CP, the TIMP score has also been associated with longitudinal cognitive, motor, and language outcomes [41]. The TIMP could be uniquely important for guiding the referral of PT services—a less consistently utilized profession in this sample—with content expertise in motor development, motor disorders, and movement therapies. The lack of relationship seen in this study could also be related to the earlier onset of PT in the NICU prior to the baseline for this study. Thus the family may have already received training and information for PT clinically.

The number of staff per bed in a NICU can vary depending on the level of NICU and location of the NICU within the U.S. [42]. The adequate number of full-time therapists in a Level III/IV NICU with high acuity can be determined via a formula developed by Craig and Smith [43]. Ross et al. [32] reported a level of adequate coverage of a high acuity Level IV NICU according to this formula. In a national survey of NICUs, 97% of Level-IV NICUs and 83% of Level-III NICUs reported having dedicated therapy teams [42]. Based on this previous work, it may be feasible to increase therapy services for children who need it, particularly OT and PT services given that ST sessions occurred most frequently within three weeks of discharge in our sample.

### Limitations

This study had some limitations which should be considered. Data were collected manually through weekly review of the medical record, and it is possible that we may have missed valuable information that was not documented in the medical record (e.g., therapist speaking with parents at a non-scheduled visit or reasons for missed therapy visits such as medical instability). Further, we did not document therapy coverage for the NICUs (i.e., how many therapists per discipline provide services), or the individual hospital distribution of roles and responsibilities between therapy disciplines. Moreover, we did not collect the recommended frequency of therapy services by each discipline, and staffing may have impacted the actual frequency, which may not have been consistent with the recommended amount in each case. This study also did not track parent presence in therapy sessions, which impacts the efficacy and carryover of therapy services. We did not record the duration of therapy sessions, or exactly what a therapist did in any given session. Data were not collected regarding the timing to full oral feeds, which is likely highly related to the need and timing of therapy services. These data may not be representative of other types of NICU settings or for samples with different socio-demographic compositions. Future work may compare referral practices to elucidate optimal referral protocols for the best service outcomes. Our work highlights that improvement is still needed in utilizing medical and developmental risk factors, as well as outcomes from therapy assessments, as the basis for referral for therapy services in the NICU.

## 5. Conclusions

This study found that services provided in the NICU were significantly different by type, but the magnitude and significance of the difference depended upon the time since enrollment week. Our study adds to previous research by demonstrating that an abnormal GMA (a predictor of high risk for developing cerebral palsy) is associated with more therapy services in the NICU, and that NMI and TIMP scores are not. These results highlight the lack of valuable clinical and developmental information being used to guide therapy referrals. The clinical impact of this work recommends that medical and developmental risk factors, as well as outcomes from therapy assessments, should be the basis for referral for therapy services in the neonatal intensive care unit.

## Figures and Tables

**Figure 1 behavsci-13-00481-f001:**
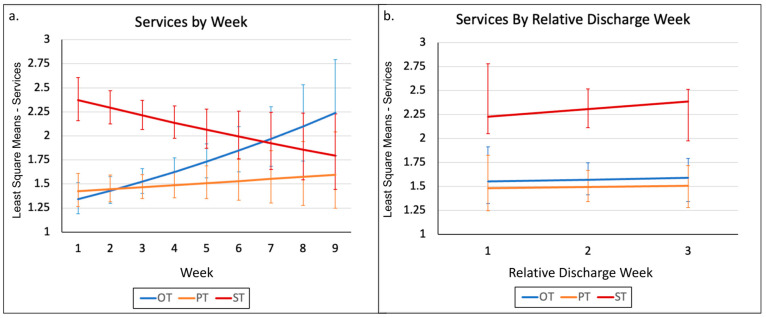
Service Types Over Time. Caption: Plots of the least square (LS) means for services over time. (**a**) A plot of least square means for the generalized linear model including an effect for service Type (OT, PT, and ST) and Week (1 to 9). (**b**) A plot of least square means for the generalized linear model including an effect for service Type (OT, PT, and ST), only on the week of discharge and the three weeks before discharge. OT = occupational therapy; PT = physical therapy; and ST = speech therapy.

**Table 1 behavsci-13-00481-t001:** Descriptives.

	Total (n = 83)	Usual Care (n = 27)	SPEEDI—Early (n = 27)	SPEEDI—Late (n = 29)
High-Risk Strata	65% (54)	63% (17)	67% (18)	66% (19)
Low-Risk Strata	35% (29)	37% (10)	33% (9)	34% (10)
Gender (Male)	46% (38)	52% (14)	44% (12)	41% (12)
Race Asian Black/AA White Multiple Unknown/Not Reported	1% (1) 31% (26) 52% (43) 14% (12) 1% (1)	4% (1) 33% (9) 44% (12) 19% (5) 0% (0)	0% (0) 26% (7) 63% (17) 11% (3) 0% (0)	0% (0) 34% (10) 48% (14) 14% (4) 3% (1)
Ethnicity Hispanic/Latino Not Hispanic/Latino Not Reported	4% (3) 93% (77) 4% (3)	4% (1) 93% (25) 4% (1)	4% (1) 93% (25) 4% (1)	3% (1) 93% (27) 3% (1)
Gestational Age—Birth (Mean (Std))	26.49 (1.99)	25.56 (1.42)	26.89 (2.03)	26.07 (2.36)
NMI 3 4 5	18% (15) 10% (8) 72% (60)	22% (6) 15% (4) 63% (17)	19% (5) 4% (1) 77% (21)	14% (4) 10% (3) 76% (22)
PSI (Mean (Std)) ^1^	64.05 (15.96)	63.17 (14.61)	62.39 (16.56)	66.24 (17.04)

Caption: Sample Descriptive Statistics. (^1^) Twenty-six (26) infants missing PSI score at baseline (9 in control, 9 in SPEEDI—Early, 8 in SPEEDI—Late); NMI = Neonatal Medical Index; PSI = Parent Stress Index—Short Form; Std = standard deviation; AA = African American; and SPEEDI = Supporting Play Exploration and Early Developmental Intervention.

**Table 2 behavsci-13-00481-t002:** Services Count All Weeks—Week of Discharge Removed.

	OT Visits		PT Visits		ST Visits		All Services	
	N	Mean (Std)	n	Mean (Std)	n	Mean (Std)	n	Mean (Std)
Week 1 Post Enrollment	73 *	1.16 (0.99)	73	1.27 (0.96)	71	1.93 (1.28)	73	4.37 (2.19)
Week 2 Post Enrollment	51	1.37 (0.94)	51	1.45 (0.97)	51	2.75 (1.13)	51	5.57 (1.66)
Week 3 Post Enrollment	38	1.76 (1)	38	1.68 (0.9)	38	2.58 (1.24)	38	6.03 (1.99)
Week 4 Post Enrollment	28	1.96 (1.07)	28	1.71 (1.12)	28	2.29 (1.3)	28	5.96 (1.75)
Week 5 Post Enrollment	18	2.00 (0.77)	18	1.67 (0.84)	18	2.17 (0.92)	18	5.83 (1.54)
Week 6 Post Enrollment	13	2.00 (1.08)	13	1.62 (1.04)	13	1.77 (1.17)	13	5.38 (1.8)
Week 7 Post Enrollment	11	1.91 (0.94)	11	1.45 (0.69)	11	1.91 (1.38)	11	5.27 (2.69)
Week 8 Post Enrollment	9	1.89 (1.45)	9	0.89 (1.05)	9	1.00 (0.5)	9	3.78 (2.33)
Week 9 Post Enrollment	4	0.50 (0.58)	4	1.50 (0.58)	4	1.75 (1.26)	4	3.75 (2.06)

Caption: Services Count All Weeks—With Week of Discharge Removed. OT = occupational therapy; PT = physical therapy; ST = speech therapy; Std = standard deviation. * A total of 10 babies were discharged within 24 h of baseline assessment and had no opportunity for therapy services in the NICU.

**Table 3 behavsci-13-00481-t003:** Services During Week of Discharge and Three Prior Weeks.

	OT Visits		PT Visits		ST Visits	
	n	Mean (Std)	n	Mean (Std)	n	Mean (Std)
3 Weeks Prior to Discharge	38	1.55 (1.01)	38	1.47 (0.95)	38	2.37 (1.17)
2 Weeks Prior to Discharge	51	1.63 (1.17)	51	1.55 (1.08)	51	2.33 (1.34)
1 Week Prior to Discharge	73	1.53 (1.11)	73	1.47 (0.9)	73	2.22 (1.2)

Caption: To specifically explore the weeks leading up to hospital discharge, we fit a generalized linear model to the data for the three weeks prior to discharge, utilizing a Poisson distribution to model the mean number of services. OT = occupational therapy; PT = physical therapy; ST = speech therapy; and Std = standard deviation.

**Table 4 behavsci-13-00481-t004:** Final Poisson Mixed Model Results.

Effect	Numerator d.f.	Denominator d.f.	F-Statistic	*p*-Value	Adjusted Effect Size Cohen’s f (95% CI)
Service Type	2	726	29.41	<0.0001 **	0.279 (0.219, 0.344)
Week	1	726	0.33	0.5673	0 (0, 0.081)
Service Type*Week	2	726	7.79	0.0005 **	0.136 (0.078, 0.203)
GMA	1	726	4.88	0.0276	0.073 (0.02, 0.143)
Baseline NMI Score	1	726	2.09	0.1491	0.039 (0, 0.115)
Caucasian	1	726	5.21	0.0228 *	0.076 (0.023, 0.146)

Caption: To evaluate if sociodemographic, neurological function, or medical risk factors, which can be measured by the medical team who make the referrals, influenced access to therapy in the NICU, we refit the initial service model (“Service Type Over Time”) described above and added in fixed effects for race (Caucasian yes/no), baseline NMI, and GMA (normal/abnormal). * = *p* < 0.05; ** = *p* < 0.001. d.f = degrees of freedom; and CI = confidence interval.

## Data Availability

The data presented in this study are available on request from the corresponding author. The data are not publicly available due to ongoing parent clinical trial.

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
