# Peer review of "Factors Influencing Receipt and Type of Therapy Services in the NICU"

_behavsci, 2023, doi:10.3390/bs13060481_

Round 1

Reviewer 1 Report

This is an interesting paper, that provides valuable insight into therapy in NICU. As highlighted by the authors, this study adds more evidence to the findings made by Ross et al (2017), and therefore is important.

Specific issues: 

1.     Legends: throughout these need to be expanded to aid the reader

2.     Although Tables and Figures are cited in the text, this needs to be improved

3.     Not all data appears to be presented in figures- for example Section 3.3- this data is not in any of the preceding tables. This should be addressed

Author Response

Reviewer 1

Yes

Can be improved

Must be improved

Not applicable

Is the content succinctly described and contextualized with respect to previous and present theoretical background and empirical research (if applicable) on the topic?

(x)

( )

( )

( )

Are all the cited references relevant to the research?

(x)

( )

( )

( )

Are the research design, questions, hypotheses and methods clearly stated?

( )

(x)

( )

( )

Are the arguments and discussion of findings coherent, balanced and compelling?

(x)

( )

( )

( )

For empirical research, are the results clearly presented?

( )

(x)

( )

( )

Is the article adequately referenced?

(x)

( )

( )

( )

Are the conclusions thoroughly supported by the results presented in the article or referenced in secondary literature?

(x)

( )

( )

( )

Comments and Suggestions for Authors

This is an interesting paper, that provides valuable insight into therapy in NICU. As highlighted by the authors, this study adds more evidence to the findings made by Ross et al (2017), and therefore is important.

Specific issues: 

  1.     Legends: throughout these need to be expanded to aid the reader

Thank you for this comment, the caption in Figure 1 has been significantly expanded to aid the reader, and all listed abbreviations have now been defined in each table caption.

  1.     Although Tables and Figures are cited in the text, this needs to be improved

The tables and figures have been more frequently cited throughout the relevant sections. 

  1.   Not all data appears to be presented in figures- for example Section 3.3- this data is not in any of the preceding tables. This should be addressed

The analyses of Prediction of individual services by Severity Strata/GMA, Baseline NMI, and Baseline TIMP,  were exploratory in nature designed to address the question "Is there an association".  To avoid an overabundance of figures due to these analyses we chose to simply describe the results from the analyses.

Reviewer 2 Report

First of all, I congratulate you on a well-written study on the mentioned topic of interest. The authors describe the frequency of therapy visits for very preterm infants in NICU and the influence of each risk factor such as NMI, TIMP score, and abnormal GMA. I believe it is useful in that it provides new insights to healthcare professionals who are doing clinical work in the NICU. However, please add a description to the following remarks so as not to confuse the readers.

Major concerns:

1) This manuscript does not state the criteria by which PT, OT, and ST visit patients. For example, Table 1 shows that OT visits patients 1.16-2.00 times/week, but please mention what criteria are used to decide whether or not to visit patients on individual days.

Minor concerns:

2) In line 49: “(ADHD, brain injury,” must be “(ADHD), brain injury,”

3) In line 156: “cerebral palsy (CP; [38, 39].” must be “cerebral palsy (CP) [38, 39].”

Author Response

Reviewer 2

Yes

Can be improved

Must be improved

Not applicable

Is the content succinctly described and contextualized with respect to previous and present theoretical background and empirical research (if applicable) on the topic?

(x)

( )

( )

( )

Are all the cited references relevant to the research?

(x)

( )

( )

( )

Are the research design, questions, hypotheses and methods clearly stated?

( )

( )

(x)

( )

Are the arguments and discussion of findings coherent, balanced and compelling?

(x)

( )

( )

( )

For empirical research, are the results clearly presented?

(x)

( )

( )

( )

Is the article adequately referenced?

(x)

( )

( )

( )

Are the conclusions thoroughly supported by the results presented in the article or referenced in secondary literature?

(x)

( )

( )

( )

Comments and Suggestions for Authors

First of all, I congratulate you on a well-written study on the mentioned topic of interest. The authors describe the frequency of therapy visits for very preterm infants in NICU and the influence of each risk factor such as NMI, TIMP score, and abnormal GMA. I believe it is useful in that it provides new insights to healthcare professionals who are doing clinical work in the NICU. However, please add a description to the following remarks so as not to confuse the readers.

Major concerns:

1) This manuscript does not state the criteria by which PT, OT, and ST visit patients. For example, Table 1 shows that OT visits patients 1.16-2.00 times/week, but please mention what criteria are used to decide whether or not to visit patients on individual days.

Neither site had a standard order set so therapy visits were based on individual physician referral. This has been added in section 2.3. 

Minor concerns:

2) In line 49: “(ADHD, brain injury,” must be “(ADHD), brain injury,”

Thank you, this change has been made

3) In line 156: “cerebral palsy (CP; [38, 39].” must be “cerebral palsy (CP) [38, 39].”

Thank you, this change has been made

Reviewer 3

Yes

Can be improved

Must be improved

Not applicable

Is the content succinctly described and contextualized with respect to previous and present theoretical background and empirical research (if applicable) on the topic?

( )

(x)

( )

( )

Are all the cited references relevant to the research?

( )

(x)

( )

( )

Are the research design, questions, hypotheses and methods clearly stated?

(x)

( )

( )

( )

Are the arguments and discussion of findings coherent, balanced and compelling?

(x)

( )

( )

( )

For empirical research, are the results clearly presented?

( )

(x)

( )

( )

Is the article adequately referenced?

(x)

( )

( )

( )

Are the conclusions thoroughly supported by the results presented in the article or referenced in secondary literature?

( )

(x)

( )

( )

Comments and Suggestions for Authors

Dear Authors:

First of all, congratulations on the completion of this research. The problem addressed is relevant and has a clinical impact for patients, relatives and related health care professionals.

However, the manuscript has limitations that should be addressed before its possible publication in this Journal.

Abstract:
The use of abbreviations in this section is discouraged. Please remove them.

Reviewer 3 Report

Dear Authors:

First of all, congratulations on the completion of this research. The problem addressed is relevant and has a clinical impact for patients, relatives and related health care professionals.

However, the manuscript has limitations that should be addressed before its possible publication in this Journal.

Abstract:
The use of abbreviations in this section is discouraged. Please remove them.
Zeros as the last decimal place do not mean anything. Please remove them throughout the manuscript.

Introduction:
This section does not address a fundamental issue in the management of neonates, which is their feeding (which, in turn, conditions other factors related to care). Doi: 10.3390/children9020150.
Providing the ClinicalTrials.gov identification code in this section is inappropriate.

Methods:
Please present descriptive figures to one decimal place throughout the manuscript.
Statistical treatment of data should be conveyed with greater clarity and brevity. In addition, statistical tests should be supplemented with other techniques such as effect size calculation.

Results:
The Tables should be redesigned in relation to their format and layout to improve and facilitate their readability. In addition, the authors should review the adequate use and explanation of abbreviations in the tables.

Discussion:
Please separate the Conclusions and state in that section the scientific novelty contributed by this research as well as its clinical impact.
Please honestly acknowledge the limitations of the research at the end of the Discussion.

Kind regards

Author Response

Reviewer 3

Yes

Can be improved

Must be improved

Not applicable

Is the content succinctly described and contextualized with respect to previous and present theoretical background and empirical research (if applicable) on the topic?

( )

(x)

( )

( )

Are all the cited references relevant to the research?

( )

(x)

( )

( )

Are the research design, questions, hypotheses and methods clearly stated?

(x)

( )

( )

( )

Are the arguments and discussion of findings coherent, balanced and compelling?

(x)

( )

( )

( )

For empirical research, are the results clearly presented?

( )

(x)

( )

( )

Is the article adequately referenced?

(x)

( )

( )

( )

Are the conclusions thoroughly supported by the results presented in the article or referenced in secondary literature?

( )

(x)

( )

( )

Comments and Suggestions for Authors

Dear Authors:

First of all, congratulations on the completion of this research. The problem addressed is relevant and has a clinical impact for patients, relatives and related health care professionals.

However, the manuscript has limitations that should be addressed before its possible publication in this Journal.

Abstract:
The use of abbreviations in this section is discouraged. Please remove them.

All abbreviations have been removed from the abstract. 

Zeros as the last decimal place do not mean anything. Please remove them throughout the manuscript.
Zeros in the last decimal place have been removed.

Introduction:
This section does not address a fundamental issue in the management of neonates, which is their feeding (which, in turn, conditions other factors related to care). Doi: 10.3390/children9020150. Providing the ClinicalTrials.gov identification code in this section is inappropriate.

We have included feeding in line 58 and line 64 to represent it as a need for babies, and an intended outcome of intervention. We’ve also added a line to the limitations regarding this (line 349-351) “Data was not collected regarding timing to full oral feeds which likely is highly related to the need and timing of therapy services.”

The clinical trials identification code has been removed from the abstract and introduction.

Methods:
Please present descriptive figures to one decimal place throughout the manuscript.
Statistical treatment of data should be conveyed with greater clarity and brevity. In addition, statistical tests should be supplemented with other techniques such as effect size calculation.

Thank you for your comment, in the appropriate place, effect sizes have been added to Table 4. Zeros in the last decimal place have all been removed. We are unable to simplify or reduce the statistical analysis text given the need to accurately include the specific parameters used for each model.

Results:
The Tables should be redesigned in relation to their format and layout to improve and facilitate their readability. In addition, the authors should review the adequate use and explanation of abbreviations in the tables.

Our team believes our tables are adequately clear and readable. We will leave this issue to the editor’s discretion. All listed abbreviations have now been defined in each table caption.

Discussion:
Please separate the Conclusions and state in that section the scientific novelty contributed by this research as well as its clinical impact. Please honestly acknowledge the limitations of the research at the end of the Discussion.

Thank you. We’ve highlighted the limitations with a new heading to designate where these start at the end of the discussion. The discussion and conclusion sections have been separated into two sections. The conclusions now state scientific novelty and clinical impact of this work. 

Round 2

Reviewer 2 Report

Revised as requested. Congratulations.

Reviewer 3 Report

Dear Authors,

Congratulations on the changes made to the manuscript. The improvements made have increased the scientific and formal quality of the manuscript.

Kind regards